# Kidney Involvement in Systemic Sclerosis

**DOI:** 10.3390/jpm12071123

**Published:** 2022-07-10

**Authors:** Francesco Reggiani, Gabriella Moroni, Claudio Ponticelli

**Affiliations:** 1Nephrology and Dialysis Unit, IRCCS Humanitas Research Hospital, Via Manzoni 56, Rozzano, 20089 Milan, Italy; gabriella.moroni@humanitas.it; 2Department of Biomedical Sciences, Humanitas University, Via Rita Levi Montalcini 4, Pieve Emanuele, 20090 Milan, Italy; 3Independent Investigator, 20131 Milan, Italy; ponticelli.claudio@gmail.com

**Keywords:** systemic sclerosis, scleroderma renal crisis, kidney involvement

## Abstract

Background: Systemic sclerosis is a chronic multisystem autoimmune disease, characterized by diffuse fibrosis and abnormalities of microcirculation and small arterioles in the skin, joints and visceral organs. Material and Methods: We searched for the relevant articles on systemic sclerosis and kidney involvement in systemic sclerosis in the NIH library of medicine, transplant, rheumatologic and nephrological journals. Results: Half of patients with systemic sclerosis have clinical evidence of kidney involvement. Scleroderma renal crisis represents the most specific and serious renal event associated with this condition. It is characterized by an abrupt onset of moderate to marked hypertension and kidney failure. Early and aggressive treatment is mandatory to prevent irreversible organ damage and death. The advent of ACE-inhibitors revolutionized the management of scleroderma renal crisis. However, the outcomes of this serious complication are still poor, and between 20 to 50% of patients progress to end stage renal disease. Conclusions: Scleroderma renal crisis still represents a serious and life-threatening event. Thus, further studies on its prevention and on new therapeutic strategies should be encouraged.

## 1. Introduction

Systemic sclerosis (SSc), also called scleroderma from its typical skin involvement, is a chronic multisystem autoimmune disease, characterized by diffuse fibrosis and abnormalities of microcirculation and small arterioles in the skin, joints and visceral organs, including the gastrointestinal tract, lungs, heart, and kidneys [1]. SSc primarily affects women and has a worldwide distribution, but its prevalence appears to be higher in North America and Australia compared to Europe and Japan [2]. The reported prevalence of SSc was between 7.2 and 33.9 per 100,000 individuals in Europe and between 13.5 and 44.3 per 100,000 in North America [3]. Women are affected more frequently than men, with the female to male ratio ranging from 3:1 to 8:1, but the factors that are responsible for this are still unknown [4]. The first manifestations may occur between the third and the fifth decade of age but tend to increase in advanced age. The rate of hospital admission, the occurrence of Raynaud’s phenomenon and death rates are higher in rainy and cold seasons [5,6]. SSc is classified upon the extent of skin alteration and the accompanying internal organs involvement [1]. Diffuse cutaneous systemic sclerosis (dcSSc) is characterized by the rapid development of symmetric skin thickening of proximal and distal extremities, face, and trunk. Patients with dcSSc are more likely to develop lung fibrosis and have an increased risk of renal crisis and other organs involvement [7]. It may also occur in association with other connective tissue diseases. The terms overlap syndrome and mixed connective tissue disease have been used to design these associations. Limited cutaneous systemic sclerosis (lcSSc) is usually limited to fingers, distal extremities, and face, while the trunk and proximal extremities are typically spared. This subset may be also associated with calcinosis, Raynaud’s phenomenon, esophageal dysmotility, sclerodactyly and telangiectasia, the so-called CREST syndrome [7].

### Pathogenesis of Systemic Sclerosis

The etiology of SSc remains incompletely understood. Genetic studies have supported the opinion that SSc patients are genetically predisposed to this disease. The strongest genetic association for SSc lies within the MHC region, with loci in HLA-DRB1, HLA-DQB1, HLA-DPB1, and HLA-DOA1 being the most replicated [8]. Also non-HLA genes are associated with SSc. Among these genes, some are involved in innate immunity, as interferon regulatory factor 5 and 7 (IRF5, IRF7) and toll-like receptor 2 (TLR2). Other genes are involved in T and B cells activation, as cluster of differentiation 47 (CD247), tumor necrosis factor alpha-induced protein 3 (TNFAIP3), signal transducer and activator of transcription 4 (STAT4) and B cell lymphocyte kinase (BLK) [9]. Single cell RNA sequencing, differential gene expression and pathway analysis revealed that endothelial cells from SSc patients show a pattern of gene expression associated with vascular injury and activation, extracellular matrix generation and negative regulation of angiogenesis. Among several genes involved in vascular injury, extracellular matrix generation and angiogenesis negative regulation, apelin receptor and heparan sulfate proteoglycan 2 have been reported to have a pivotal role in endothelial cell injury [10]. Not only genetic predisposition but also environmental factors and epigenetic influences contribute to the development of SSc. In a meta-analysis aimed to assess whether scleroderma was associated with occupational and environmental exposure, silica and organic solvents were the two most likely substances related to the development of scleroderma [11].

The pathogenic mechanism that brings to the development of SSc in a genetically predisposed subject remains elusive. However, early morphologic endothelial cell injuries, which can be seen before the disease develops, suggest that SSc pathogenesis starts in the vasculature (Figure 1) [12]. Thus, in a predisposed genetic background, the blood vessels modifications, probably caused by exogenous triggers, may engage inflammatory molecules and cells of the innate immune system. In this setting there is a crosstalk between inflammatory cells and fibroblasts leading to increased myofibroblasts transdifferentiation, that can produce pro-fibrotic mediators, including transforming growth factor-beta (TGF-β) and interleukin 13 (IL-13) [13,14]. In the inflammatory environment several antigens are produced and dendritic cells intercept them, become mature, migrate to lymph nodes, and present the antigens to T cells, activating adaptive immunity through the costimulatory CD40L–CD40 axis. Once activated by the contact and costimulation with the antigen, T cells release interleukin 2 (IL-2), proliferate and differentiate into effector T cells [15]. However, impaired function of regulator T cells and B cells hyperactivity lead to important alterations of the adaptive immunity [16,17]. The upregulated B cells lead to the production of a plethora of autoantibodies, such as anti-topoisomerase I and anti-centromere antibodies. While anti-topoisomerase I antibodies are usually associated with diffuse and severe SSc [18,19,20], anti-centromere antibodies are associated with the CREST syndrome or lcSSc [21,22]. Among other autoantibodies, those targeting endothelial cells, including endothelin type A receptor and angiotensin II type I receptor, induce activation or apoptosis of endothelial cells [23].

Activation of autoantibodies and cell-mediated autoimmunity, through the mediation of endothelin 1 (ET-1), may also lead to endothelial dysfunction and development of microvascular endothelial cells/small vessels fibroproliferative vasculopathy, with subsequently chronic ischemia of the affected tissues [24]. ET-1 is a potent vasoconstrictor and profibrotic factor mainly produced by endothelial cells in response to several stimuli, as physical stress, cytokines, growth factors, and thrombin. ET-1 decreases the production of the vasodilator nitric oxide and induces oxidative stress and fibrosis. Of note, HLA-B35 influences the production of ET-1 [25]. 

In this permissive genetic background, abnormalities of the innate and adaptive immune systems, coupled with increased expression of ET-1 [26] and reduced production of nitric oxide synthase [27], cause vasoconstriction, intimal proliferation and hypoxia with consequent endothelial-mesenchymal transition and conversion to fibroblast-like cells [28]. In parallel, ET-1 activates and re-programs the functional phenotypes of vascular smooth muscle cells, microvascular pericytes and tissue fibroblasts into pro-fibrogenic cell populations with myofibroblasts-like properties [29]. An important role in promoting collagen release by fibroblasts is played by TGF-β. TGF-β activates autophagy by an epigenetic mechanism to amplify its profibrotic effects. In fact, activation of autophagy in fibroblasts promotes collagen release and is sufficient and required to induce tissue fibrosis [30].

Also platelets are activated in SSc and can release chemokines, cytokines and growth factors, including transforming growth factor-β1 (TGF-β1), which increases the production of collagen and extracellular matrix [31]. In the development and progression of fibrosis in SSc patients the platelet-derived growth factor (PDGF)/platelet-derived growth factor receptor (PDGFR) pathway has an important role. PDGF and PDGFR resulted to be upregulated in SSc and this induces fibrosis and myofibroblasts differentiation [32]. Moreover, in SSc autoantibodies against PDGFR have been identified. These autoantibodies activate smooth muscle cells and contribute to the development of SSc intimal hyperplasia [33], and tissue fibrosis [34].

Thus, the interplay between immune components, inflammatory processes and vascular changes generates a population of activated connective tissue-producing fibroblasts and myofibroblasts, eventually causing excessive accumulation of collagen and other matrix components in skin, joints, and visceral organs [35].

## 2. Kidney Involvement in Systemic Sclerosis

Kidney disease in SSc is common, as demonstrated by autopsy studies, in which renal histological involvement was present in 60 to 80 percent of patients [36,37]. Scleroderma renal crisis (SRC), which occurs in a minority of patients, is the most serious complication. Kidney involvement is more frequently observed in the context of extensive diffuse skin disease and typically occurs in the first two years, simultaneously with the worsening of the skin involvement [38].

Beyond SRC, some patients may show mild hypertension and normal renal function. These patients are initially asymptomatic or show only mild proteinuria, microscopic hematuria and occasional casts. Nephrotic syndrome is uncommon. Mild kidney dysfunction may be also common, as demonstrated by Iliopoulos et al., which examining 796 SSc patients from five different studies observed that 31.5% had a glomerular filtration rate (GFR) < 90 mL/min and 19.5% a GFR < 60 mL/min [39]. Patients with kidney dysfunction usually show tissue fibrosis and vasculopathy at renal biopsy [40]. These alterations are usually not progressive and kidney dysfunction tends to decline at a rate similar to the general population [38,39]. Progression to end-stage renal disease (ESRD) may be accelerated by the presence of arterial hypertension and moderate proteinuria [41]. Despite the slow progression of kidney dysfunction, there is a strong association between kidney involvement and outcomes in SSc, with a threefold increased risk of mortality from pulmonary hypertension if renal insufficiency is present [42].

There may be also positivity for anti-dsDNA antibodies or antineutrophil cytoplasmic antibodies (ANCA), which may indicate a possible evolution to another connective disease even in the absence of clinical changes [43,44,45]. Anti-dsDNA antibodies may be found both in lcSSc and dcSSC [46], but their clinical significance is unknown. ANCA have been found to be associated with interstitial lung disease and pulmonary embolism [47].

### 2.1. Scleroderma Renal Crisis

SRC may occur in about 10% of all patients with scleroderma [48], and it’s more frequently observed in patients with dcSSc [49]. SRC invariably develops within the first 3–5 years after the onset of scleroderma signs or symptoms, but sometimes SRC precedes the clinical diagnosis of SSc [50,51]. 

SRC typically presents with an abrupt onset of moderate to marked hypertension and kidney failure without signs of glomerulonephritis [52]. Hypertension is frequently accompanied by manifestations of malignant hypertension, as headache, blurred vision and convulsions, often followed by acute heart failure, pulmonary edema, and renal failure. The onset of kidney failure is acute and usually in the absence of significant previous kidney involvement. The urine sediment is usually normal, cellular casts are uncommon and proteinuria may be present, but usually is less than 1 g per day and present in patients with concomitant hypertension [41]. In addition to this clinical presentation, also microangiopathic hemolytic anemia with low platelets may occur in about half of the cases [53]. SRC may have a dramatic clinical presentation and represents an emergency since without intervention permanent kidney failure or even death may occur.

Several risk factors for SRC have been identified. These include the presence of a diffuse skin involvement, the presence or absence of some serum autoantibodies and the use of certain drugs, as glucocorticoids and cyclosporine (CYC). In a large retrospective study, the following variables were significantly associated with the risk of SRC: proteinuria, anemia, hypertension, chronic kidney disease, elevated erythrocyte sedimentation rate, thrombocytopenia, hypothyroidism, anti-Ro antibody and anti-RNA-Polymerase III antibody seropositivity. Three or more of these risk factors present at SSc diagnosis were sensitive (77%) and highly specific (97%) for future SRC [54]. The most important risk factor for SRC appears to be diffuse skin involvement, in particular if it is rapidly progressive [50,55]. Other reports pointed out the frequent presence of anti-RNA-Polymerase III antibodies in patients who developed SRC [19,56]. Anti-topoisomerase1 antibodies are instead associated with disease progression and poor outcome of SRC [20,57]. The use of high doses of corticosteroids has been associated with an increased risk of SRC, as demonstrated by several retrospective studies [49,50,58]. The possibility that CYC may induce SRC is more anecdotal and not confirmed by solid evidence.

In around 15% of cases, SRC may present with normal blood pressure and is termed normotensive SRC [51]. The initial onset may be insidious until the presence of hemolytic anemia and thrombocytopenia heralds the development of thrombotic microangiopathy, which is often followed by severe hypertension and oliguric renal failure [59]. The differential diagnosis with other forms of thrombotic microangiopathy may be difficult. However, in thrombotic microangiopathy secondary to SRC, the levels of the metallopeptidase with thrombospondin type 1 motif 3 enzyme (ADAMTS13) and complement factor H are usually normal [60]. Moreover, patients with other forms of thrombotic microangiopathy have no clinical or serological signs of SSc.

### 2.2. Pathogenesis of Scleroderma Renal Crisis

Although the pathogenesis of SRC is not yet fully elucidated, it is likely that both endothelial dysfunction and vascular damage concur to produce a series of insults to the kidneys resulting in vasospasm and narrowing of renal arterioles. The frequent presence of anti-RNA-Polymerase III antibodies and the previous exposure to glucocorticoids makes it possible to speculate that exogenous stimuli may induce vasoconstriction of renal arterioles in predisposed patients with SSc. ET-1 may represent the trigger since its overexpression has been shown in glomeruli and arterioles of kidney biopsies of patients with SRC [61,62]. ET-1 is the most potent vasoconstrictor in the human body [63]. Powerful vasoconstriction modifies the vascular tone, contributing to tissue hypoxia. ET-1 can also re-program the vascular smooth muscle cells, microvascular pericytes and tissue fibroblasts into pro-fibrogenic cell populations with myofibroblasts-like properties [29]. The vascular damage may be further aggravated by the overactivation of vasoconstrictor angiotensin II and rapid degradation of vasodilator bradykinin caused by the angiotensin-converting enzyme (ACE) I/D polymorphism, which is particularly frequent in SSc [64]. After the endothelial injury, there is a cascade of histologic alterations that starts with a rapid increase in endothelial permeability and intimal edema. This causes a direct contact of the subendothelial connective tissue with circulating blood elements activating the coagulation cascade and vascular thrombosis. The connective tissue reacts to this insult by promoting fibroblastic and non-fibroblastic stromal proliferation, causing proliferative endo arteriopathy. Decreased renal perfusion caused by arterial narrowing leads to juxtaglomerular apparatus hyperplasia and renin secretion, resulting in accelerated hypertension and progressive renal injury [65]. The critical rise in blood pressure not only causes further damage to kidney blood vessels, but also initiates a vicious cycle eventually leading to malignant hypertension and acute renal failure.

### 2.3. Histologic Changes of Scleroderma Renal Crisis

The histologic changes are seen characteristically in arcuate, interlobular and small arteries, and arterioles (Figure 2). 

The early changes consist in proliferation of intimal cells and accumulation of mucopolysaccharides and glycoproteins. This proliferation damages the internal elastic lamina allowing muscle-like cells to migrate into the intima. A fibrous thickening of adventitia also occurs. Gross narrowing of the blood vessels leads to ischemia which may be aggravated by vascular thrombosis, with consequent atrophy of the nourished tissues. Fibrinoid changes in the walls of arterioles and microinfarcts are also frequent [66,67]. Glomeruli may be normal but may show areas of fibrin deposits (Figure 3).

In the frequent cases of thrombotic microangiopathy, there are mesangiolysis, thickening of capillary walls, intracapillary thrombosis and areas of fibrinoid necrosis. Juxtaglomerular apparatus is hypertrophic. Immunofluorescence studies are non-specific with occasional glomerular and vascular deposits of IgM, C3 and fibrin. Electron dense deposits are not seen. The tubules may show changes due to ischemic injury with tubular degeneration and necrosis in acute stages. When the damage becomes chronic, tubular atrophy and interstitial fibrosis develop and are proportional to vascular injury [68]. Both post- and ante-mortem studies suggest that endothelial lesions occur before the clinical diagnosis of renal disease and precede the histological evidence of fibrosis [69]. 

SSc may overlap with lupus nephritis [70], while the association with other glomerular diseases, as IgA nephropathy, IgM nephropathy, and membranoproliferative glomerulonephritis is infrequent [71]. Additionally, the possible association with ANCA positivity has been described, although uncommon (0–12%) and only in rare cases with clinical characteristics of ANCA-associated vasculitis [72]. The possible association with other diseases emphasizes the role of kidney biopsy in confirming the clinical diagnosis, predicting the clinical outcome, and optimizing patient management [65].

### 2.4. Prognosis of Scleroderma Renal Crisis

The prognosis of scleroderma renal crisis largely depends on the rapid control of malignant hypertension and improvement of the ongoing renal ischemia. The advent of angiotensin-converting enzyme inhibitors (ACEIs) allows for the reversal of the unfavorable outcome of SRC. In a pivotal prospective study carried out in a cohort of 108 SRC patients, the 1 year survival rate was 76% in those treated with ACEIs and 15% in those who were not in treatment with ACEIs [73]. Despite the improvement, SRC remains a life-threatening complication characterized by a high rate of mortality and progression to ESRD and dialysis. In a retrospective study, the clinical charts of 606 patients affected with SSc were reviewed. 20 (3.3%) patients developed SRC. One year after SRC onset, 11 (55%) patients developed ESRD. The survival rate was 70% at 1 year and 50% at 5 years [74]. In another retrospective study, 49 of 91 patients with SRC (53.8%) required dialysis, which was definitive for 38. 37 (40.7%) SRC patients died. Death was most frequent among dialyzed patients who never recovered renal function (22 vs. 2) and 13 never-dialyzed SRC patients died [52]. In summary, short-term prognosis of SRC has improved, but long-term prognosis remains disappointing, particularly in patients in renal replacement therapy [49,75]. Mortality is reduced but a substantial number of patients with SRC still die of cardiac or lung complications [76,77].

## 3. Treatment

SSc may present with a wide spectrum of manifestations. Thus, the treatment should be tailored to the type of organ involvement and to the disease subset (Table 1).

Often patients are treated with symptomatic therapy for the specific organ involvement, as proton pump inhibitors for gastrointestinal reflux or calcium channel blockers (CCBs) for Raynaud phenomenon. Immunosuppressive drugs, such as methotrexate, azathioprine, mycophenolate and cyclophosphamide (CYC), are used to manage interstitial lung disease, which remains the leading cause of mortality in SSc [78,79,80]. However, immunosuppression is not curative [1], and some drugs, such as corticosteroids and maybe CYC, should be avoided in patients with SSc, as they can favor the development of SRC [81,82]. It has been reported that the risk to develop SRC increased by 1.5% for every mg of prednisone/day consumed the trimester prior to SRC [83]. Rituximab was associated with a good safety profile in a large observational SSc cohort. Significant change was observed on skin fibrosis, but not on lung involvement [84]. Bosentan, a dual endothelin-receptor antagonist, has been shown to be useful for the treatment of pulmonary arterial hypertension and to prevent new digital ulcers [85,86]. Sildenafil may reduce proinflammatory activation induced by oxidative stress and can improve the microvascular blood flow in patients with SSc [87]. Imatinib mesylate has been shown to inhibit the profibrotic activity of TGF-β and to prevent lung fibrosis in a mouse model [88]. A systematic review reported that after a treatment period of 6 to 12 months, imatinib mesylate significantly improved the skin score, whereas health-related assessment questionnaires remained unchanged. Data regarding change in pulmonary function tests were insufficiently consistent. Regarding safety, the authors found a pooled dropout rate due to all adverse events of 22% and a rate of serious adverse events of 17% [89]. Nintedanib, a tyrosine kinase inhibitor with anti-fibrotic and anti-inflammatory effects, is effective in reducing the effects of interstitial lung involvement. In a randomized controlled trial, patients with SSc assigned to receive nintedanib had a significantly lower annual decline in forced vital capacity compared to those assigned to placebo [90]. Experimental studies have shown that rapamycin, specific inhibitor of mTOR, inhibits TGF-β1 induced myofibroblast differentiation, extracellular matrix production, and collagen contraction [91]. A small randomized trial in patients with SSc showed that the efficacy of rapamycin was similar to methotrexate [92].

In SSc patients with chronic kidney disease, a careful check and treatment of arterial hypertension is recommended [93]. Treatment should be started very early, before irreversible kidney disease occurs. ACEIs and/or angiotensin II receptor blockers (ARBs) are the drugs more frequently used to control blood pressure in SSc. They should be used to maximally tolerate or allowed as a dose as the first-line in treating patients with both hypertension and proteinuria. Calcium channel blockers are also used to treat hypertension, reduce the vasospasm induced by renal vasoconstriction and treat Raynaud phenomenon. ACEis, ARBs and CCBs should not be used to prevent SRC occurrence. There are also some concerns about the use of ACEIs in patients at high risk of developing SRC, despite strong evidence supporting these concerns is lacking. In a large observational cohort of over 14,000 patients with SSc followed for five years, an increased risk of developing SRC (HR 2.6, 95% 1.7 to 4.0) was associated with the use of ACEIs [94]. In a retrospective cohort was found that ACEIs use at SSc diagnosis was associated with an increased risk for SRC, but only when associated with the presence of proteinuria, suggesting that ACEis use may be a passive marker of known SRC risk factors [95].

Despite this, ACEIs are still considered as the first choice in SRC treatment, but their use should be more closely monitored [94,95]. In case of SRC, early and aggressive treatment is mandatory to prevent irreversible organ damage and death [36]. However, blood pressure should not be reduced too rapidly or below the limit of cerebral autoregulation to prevent the risk of iatrogenic ischemia. Thus, intravenous antihypertensives such as nitroprusside and labetalol, or powerful drugs such as minoxidil, should be avoided. Mean arterial pressure should be reduced by approximately 10 to 20% within the first hour and by another 5% to 15% over the next 24 h. This often results in a target blood pressure of less than 180/120 mmHg for the first hour and less than 160/110 mmHg for the next 24 h, but rarely less than 130/80 mmHg during that time frame [96,97,98]. The decision of the drug to use depends on several factors, including the clinical indications, pharmacokinetics, toxicity and drug interactions. Furthermore, two or more drugs can be required for the successful lowering of the patient’s blood pressure. ACEIs remain the mainstay in the therapy of SRC due to the critical role of RAAS in the pathogenesis of SRC. Several nonrandomized, uncontrolled, retrospective and prospective studies demonstrated that ACEIs are associated with a better control of blood pressure, improvement of kidney function, and reduction in mortality rate in patients with SRC [50,73,99,100,101,102]. The same effect has not been observed with ARBs [103]. Dihydropyridine calcium-channel blockers, such as nifedipine, can also reverse SRC [86,104]. Other therapeutic strategies have been tested; plasma-exchange seems to give some benefits, particularly in patients with SRC and microangiopathy or in subjects intolerant to ACE-inhibitors [105,106]. SRC may lead to complement system activation through the classical pathway. Early administration of eculizumab, a monoclonal antibody blocking complement component 5 (C5), has been used with success in patients with SRC complicated by thrombotic microangiopathy [107,108,109]. In refractory cases, a combination of ACEIs with endothelin receptor blockers [110], and agents targeting C5 has been proposed [111].

Even with treatment with ACEIs, about 20 to 50% of patients with SRC require dialysis [50,58,73]. Both hemodialysis and continuous ambulatory peritoneal dialysis (CAPD) have been used in patients with SSc. A review of the Australian/New Zealand data reported that 50% of dialysis patients with SSc were treated with CAPD. The cumulative median survival was significantly shorter in scleroderma patients than in other dialysis patients (2.43 years vs. 6.02 years). Scleroderma was found to be an independent predictor for mortality. However, 10% of scleroderma dialysis patient vs. 1% of other dialysis patients had renal recovery [112]. In the USA, overall ESRD rates from scleroderma were 0.5 per million per year but fell to 0.42 in 2012. After initiating renal replacement therapy, patients with scleroderma had a greater likelihood of recovery of kidney function (HR = 2.67) and death (HR = 1.44), and a lower likelihood of transplantation (HR = 0.51) than demography-matched patients without scleroderma [113]. In Europe, between 2002 and 2013, the range of adjusted annual incidence and prevalence rate of regular replacement therapy for ESRD due to scleroderma were 0.11 to 0.26 and 0.73 to 0.95 per million population, respectively. Recovery of independent kidney function in the scleroderma group was 7.6% but time required to achieve recovery was long. The 5 year survival probability among patients with scleroderma was 38.9%, lower than that observed in other groups [114].

Many patients do not recover kidney function and remain dialysis-dependent. However, a few patients may recover kidney function after a long period of dialysis. Thus, decisions regarding kidney transplantation should be taken after 1–2 years of dialysis [115]. Scleroderma recurrence after kidney transplantation is possible but the risk is low. Risk factors include progression of diffuse skin thickening, new-onset anemia and cardiac complications [116]. In the Australian and New Zealand review, 5 year deceased donor and live donor renal allograft survival rates of recipients with scleroderma were 53 and 100%, respectively [112]. The UNOS registry reported a graft survival at 1 and 3 years of 68% and 60%, respectively; significantly better than in patients with SSc remained in the waiting list. Early graft loss was common [117]. In the European review, 5 year post transplantation patient survival and 5 year allograft survival were 88.2% and 72.4% [114], respectively. A French multicenter study retrospectively reviewed the outcome of 34 patients who received 36 kidney transplants between 1987 and 2013. Extrarenal involvement of SSc was generally stable, except for cardiac and gastrointestinal involvements, which worsened after kidney transplantation in 45% and 26% of cases, respectively. Patient survival was 82.5% at 5 years post-transplant. Pulmonary involvement of SS was an independent risk factor of death after transplantation. Death-censored graft survival was 92.8% after 5 years. Recurrence of SRC was diagnosed in three cases. Based on that study, in the absence of extrarenal contraindication, SSc patients presenting with ESRD should be considered for kidney transplantation [118]. Little information is available about the optimal immunosuppression in transplanted SSc patients. Ideally, the use of corticosteroids should be minimized. Both CYC and tacrolimus are profibrotic drugs that may aggravate fibrosis in the skin, lungs and vascular system. The mTOR inhibitors may replace partially or completely the use of calcineurin inhibitors. There is no contraindication to the use of azathioprine, mycophenolate salts, belatacept or biologic agents, such as basiliximab and anti-thymocyte globulins.

## 4. Conclusions

Up to half of patients with SSc have clinical evidence of kidney involvement, such as mild proteinuria, elevated serum creatinine concentration, or hypertension. Despite the identification of possible risk factors, as the use of glucocorticoids and the presence of autoantibodies directed against RNA polymerase III, SRC develops in about 10% percent of SSc patients, most commonly within 3 to 5 years of diagnosis, and in those with dcSSC. Early and aggressive treatment is mandatory to prevent irreversible organ damage and death. The mainstay of therapy in SRC is a prompt and effective blood pressure control using ACEIs. Despite the advent of ACEIs, about 20 to 50% of patients with SRC progress to ESRD and require dialysis. However, a significant proportion of patients on renal replacement therapy will recover renal function. Kidney transplantation should be considered after a period of 1 to 2 years on dialysis. Considering the poor outcomes that still characterize SRC, further studies on its prevention and on new therapeutic strategies should be encouraged.

## Figures and Tables

**Figure 1 jpm-12-01123-f001:**
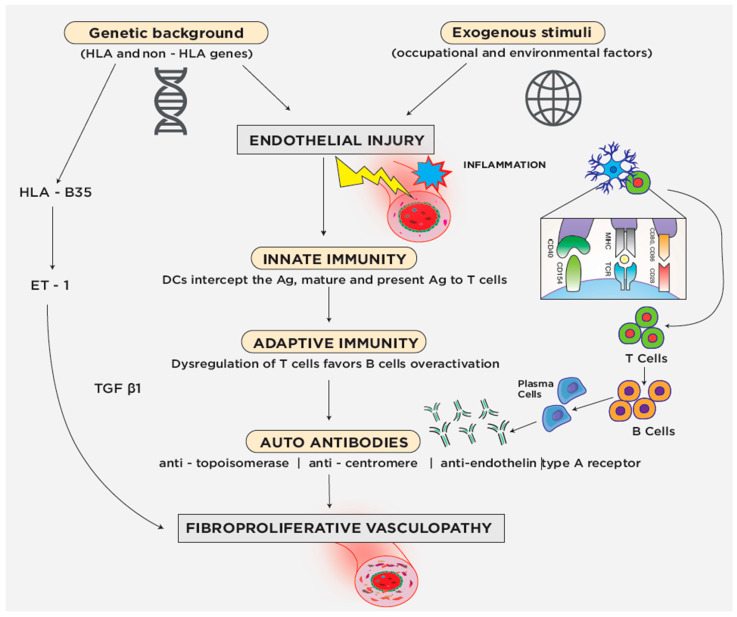
In a predisposed genetic background, an exogenous trigger that targets blood vessels may determine the engagement of inflammatory molecules and cells of the innate immune system. In the inflammatory environment, dendritic cells intercept the antigen, become mature, migrate to lymph nodes, and present the antigen to T cells, activating adaptive immunity through the costimulatory CD40L–CD40 axis. Once activated, T cells release interleukin 2, proliferate and differentiate into effector T cells. However, impaired function of regulator T cells and B cells hyperactivity lead to the production of a plethora of autoantibodies, such as anti-topoisomerase I, anti-centromere and anti-endothelin type A receptor antibodies.

**Figure 2 jpm-12-01123-f002:**
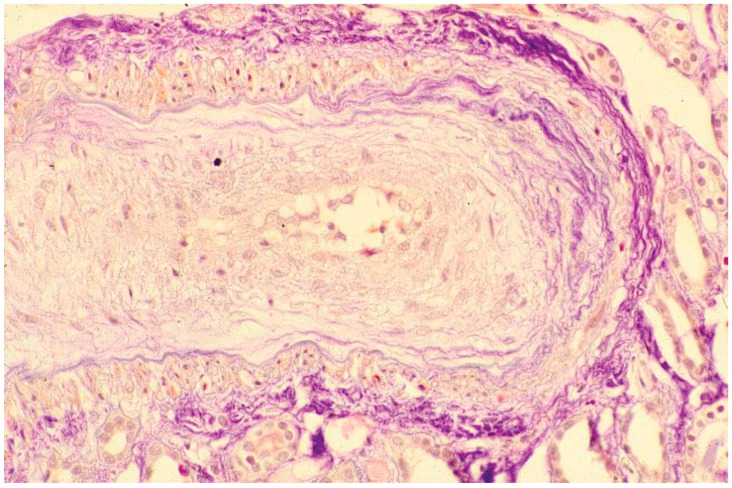
Light microscopy: section of an interlobular arterial wall that shows intimal mucoid edema, resulting in “onion skin” concentric appearance and a reduction in the lumen.

**Figure 3 jpm-12-01123-f003:**
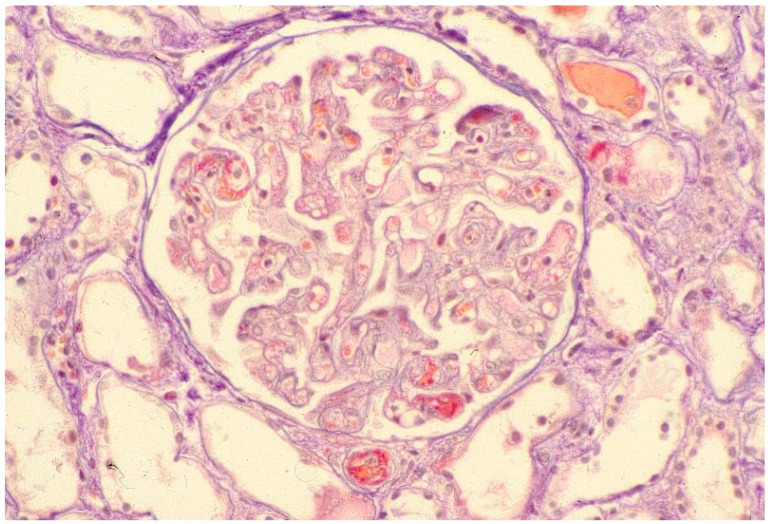
Light microscopy: ne glomerulus that shows fibrin thrombi of some capillary lumens, leukocytes infiltration, and a fibrin thrombus of the afferent arteriole.

**Table 1 jpm-12-01123-t001:** Main characteristics of drugs currently used or under investigation in systemic sclerosis (SSc). cGMP, cyclic guanosine monophosphate (cGMP); dcSSC, diffuse cutaneous systemic sclerosis; mTOR, mammalian target of rapamycin; SRC, scleroderma renal crisis.

	SSc Drugs	Mechanism of Action	Use in SSc	Concerns in SSc
Symptomatic treatment	Proton pump inhibitors	Inhibit stomach’s H+/K+ ATPase proton pump	Symptomatic treatment of gastrointestinal reflux	
	Calcium channel blockers	Vasodilation induced by calcium channel blockade	Symptomatic management of Raynaud phenomenon	
Immunomodulation	Corticosteroids	Inhibition of inflammation-associated molecules	Management of interstitial lung disease	Favor SRC development. Side effects with long-term, high-dose treatment.
	Methotrexate	Antimetabolite (inhibits dihydrofolate reductase)	Management of interstitial lung disease and dcSSc	
	Azathioprine	Inhibition of purine synthesis	Management of interstitial lung disease	
	Mycophenolate	Inhibition of inosine-5′-monophosphate dehydrogenase	Management of interstitial lung disease and dcSSc	
	Cyclophosphamide	Cell apoptosis caused by DNA crosslinks	Management of interstitial lung disease	Favor SRC development. Efficacy as a single agent is unknown.
	Rapamycin	mTOR inhibitor	Antifibrotic?	Further studies required
Antifibrotic Drugs	D-penicillamine	Interference with collagen biosynthesis	Not clearly determined	
	Iloprost	Synthetic analogue of prostacyclin	Vasodilating effect and antifibrotic	
	Nintedanib	Tyrosine kinase inhibitor	Management of interstitial lung disease	
	Pirfenidone	Inhibition of fibroblast proliferation and collagen production	Antifibrotic. Management of interstitial lung disease?	Further studies required
Products under investigation	Bosentan	Dual endothelin-receptor antagonist	Effective in pulmonary arterial hypertension, prevent new digital ulcers	
	Intravenous immunoglobulin	Not known the antifibrotic mechanism	Management of both skin and visceral involvement	
	Tocilizumab	Anti-human IL-6 receptor antibody	Antifibrotic activity in vitro	
	Sildenafil	Inhibition of cGMP-specific phosphodiesterase type 5	Improve the microvascular blood flow	
	Imatinib mesylate	Inhibition of tyrosine kinase enzymes	Prevention of lung fibrosis?	
	Rituximab	Anti-CD20 antibody	Skin fibrosis	Not effective in lung disease

## Data Availability

Not applicable.

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
