# Peer review of "Kidney Involvement in Systemic Sclerosis"

_jpm, 2022, doi:10.3390/jpm12071123_

Round 1

Reviewer 1 Report

Interesting and important topic but not innovative; reliable literature review, taking into account the latest opinions;

I did not find figures on histopathology

Author Response

Dear editor,

thank you for the useful advices.

We uploaded the figures on histopatology as you suggested. 

Best regards

Francesco Reggiani

Reviewer 2 Report

The manuscript of Reggiani et al. is a review focused mostly on the clinical aspects of systemic sclerosis and, in particular, on kidney involvement. A detailed revision of the current therapeuthic strategies is provided.

I only have minor comments.

In general, text reading is not always fluent, since sometimes sentences are disconnected and not always well organized in a sequential order. The authors should try to improve. 

Abstract - Background - "abnormalities of small arteries" AND Introduction - "small arteries abnormalities": vascular damage in SSc affects primarily the microcirculation and small arterioles. Please, correct.

Abstract - Materials and Methods - "We searched for the relevant articles": relevant articles on what topic? Please, specify.

Paragraph "Pathogenesis of SSc" - Sentence from ""There are also non-HLA genes..." to "B cell lymphocyte kinase (BLK)" is difficult to follow. Please, provide a clearer list of genes associated with SSc. 

Paragraph "Pathogenesis of SSc" - "Single cell RNA sequencing, differential gene expression and pathways analysis...": in which kind of cells, tissues or patients were these analyses performed? Please, specify.

Paragraph "Pathogenesis of SSc" - "Among several genes...": which genes?upregulated genes? Please, clarify.

Paragraph "Pathogenesis of SSc" - "Further pathways...": please, specify which pathways.

Paragraph "Pathogenesis of SSc" - "leading to overproduction of myofibroblasts": the term "overproduction" is incorrect, since myofibroblasts transdifferentiate from fibroblasts or originate from other resident cell types. Please, correct.

Paragraph "Pathogenesis of SSc" - "dendritic cells intercept the antigen": which antigen? Please, clarify.

Paragraph "Pathogenesis of SSc" - "through the mediation of endothelin 1 (ET-1)": please, clarify the role of endothelin 1. Which cells produce it? In which conditions? When it is overexpressed?

End of paragraph "Pathogenesis of SSc" - Since the authors discuss about the involvement of fibroblasts and platelets in SSc fibrosis, about the role of TGFb in promoting collagen production and tissue fibrosis, and about anti-PDGFR autoantibodies, I would suggest to mention also the role of PDGF/PDGFR in these processes (please, see Paolini C et al., IJMS 2022). PDGFR autoantibodies do not contribute only to vascular damage but also to tissue fibrosis (by activating fibroblasts) (please, see Moroncini G et al., AR 2015; Luchetti MM et al., AR 2016).

End of paragraph "Kidney involvement in SSc" - Which clinical manifestations are associated with the positivity for anti-dsDNA antibodies or ANCA antibodies? 

Paragraph "Treatment" - I would suggest to add a table with "name, mechanism of action, current clinical indications" of the different drugs which are presented; it will help to follow this part of the manuscript.   

Minor spell check

"Platelet-derived growth factor receptor (PDGFR) autoantibodies activate smooth MUSCLE cells..."

"Three or more of these risk factors present at SSc diagnosis WERE sensitive..."

"Fibrinoid changes in the walls OF arterioles..."

"Glomeruli may be normal but may be show..."

Author Response

Dear Editor,

thank you for the useful advices.

We modified the manuscript according to your suggestions, trying also to improve the fluency of the sentences.

Best regards

Francesco Reggiani